JCB Journal of Cell Biology

**REPORT**

# Mitochondrial stress causes neuronal dysfunction via an ATF4-dependent increase in L-2-hydroxyglutarate

Rachel J. Hunt[1], Lucy Granat[1], Gregory S. McElroy[2], Ramya Ranganathan[1], Navdeep S. Chandel[2], and Joseph M. Bateman[1]

**Mitochondrial stress contributes to a range of neurological diseases. Mitonuclear signaling pathways triggered by mitochondrial stress remodel cellular physiology and metabolism. How these signaling mechanisms contribute to neuronal dysfunction and disease is poorly understood. We find that mitochondrial stress in neurons activates the transcription factor ATF4 as part of the endoplasmic reticulum unfolded protein response (UPR) in *Drosophila*. We show that ATF4 activation reprograms nuclear gene expression and contributes to neuronal dysfunction. Mitochondrial stress causes an ATF4-dependent increase in the level of the metabolite L-2-hydroxyglutarate (L-2-HG) in the *Drosophila* brain. Reducing L-2-HG levels directly, by overexpressing L-2-HG dehydrogenase, improves neurological function. Modulation of L-2-HG levels by mitochondrial stress signaling therefore regulates neuronal function.**

## Introduction

Mitochondria play key roles in cellular metabolism including synthesis of ATP, amino acids, reactive oxygen species (ROS), nucleotides, heme, and cholesterol (Spinelli and Haigis, 2018). Efficient mitochondrial function is paramount in the nervous system, and mitochondrial diseases frequently result in motor dysfunction, seizures, ataxia, and intellectual disability. Compromised mitochondrial function is also a trait of common neurological diseases including neurodegenerative conditions, autism spectrum disorders, and schizophrenia (Khacho et al., 2019).

Defects in mitochondrial gene expression cause imbalance of mitochondrial/nuclear encoded oxidative phosphorylation (OXPHOS) subunits and failure to correctly assemble OXPHOS complexes. The resulting mitochondrial stress triggers mitonuclear signaling mechanisms that enable mitochondria to communicate with the nucleus and reprogram nuclear gene expression (Arnould et al., 2015). These signaling mechanisms include the mitochondrial unfolded protein response (UPR), mitochondrial proteolytic response, and heat shock response pathways. Mitochondrial stress in neurons reprograms metabolism and nuclear gene expression, potentially regulating neuronal function (Hunt and Bateman, 2018). However, the mechanisms by which mitochondrial stress regulates neuronal function are poorly understood.

## Results and discussion

### Mitochondrial stress–induced activating transcription factor 4 (ATF4) causes neuronal dysfunction

To investigate how mitochondrial stress affects the nervous system, we used inducible overexpression of the mitochondrial transcription factor TFAM in *Drosophila melanogaster*. TFAM is an HMG box DNA binding protein that binds to the mitochondrial DNA light strand promotor in a complex with the mitochondrial RNA polymerase and TFB2M (Hallberg and Larsson, 2011; Stros et al., 2007). Non-specific binding of TFAM packages mitochondrial DNA into its nucleoid structure, at high levels repressing gene expression (Kang et al., 2018; Ylikallio et al., 2010). TFAM overexpression inhibits mitochondrial OXPHOS gene expression (Cagin et al., 2015; Pohjoismaki et al., 2006; Ylikallio et al., 2010), and in the nervous system causes mitochondrial and neuronal dysfunction and triggers mitonuclear stress signaling (Cagin et al., 2015; Chen et al., 2019; Duncan et al., 2018; Tsuyama et al., 2017).

ATF4 is activated by a variety of different mitochondrial stresses in mammalian cultured cells and in vivo (Celardo et al., 2017; Quirós et al., 2017), and persistent activation can be detrimental (Khan et al., 2017). ATF4 is activated in animal models of Alzheimer's disease, amyotrophic lateral sclerosis, and frontotemporal dementia, which exhibit neuronal mitochondrial stress (Hughes and Mallucci, 2019). We find that in keeping with

[1]Maurice Wohl Clinical Neuroscience Institute, King's College London, London, UK; [2]Department of Medicine and Biochemistry and Molecular Genetics, Northwestern University Feinberg School of Medicine, Chicago, IL.

Correspondence to Joseph M. Bateman: joseph_matthew.bateman@kcl.ac.uk; Navdeep S. Chandel: nav@northwestern.edu.

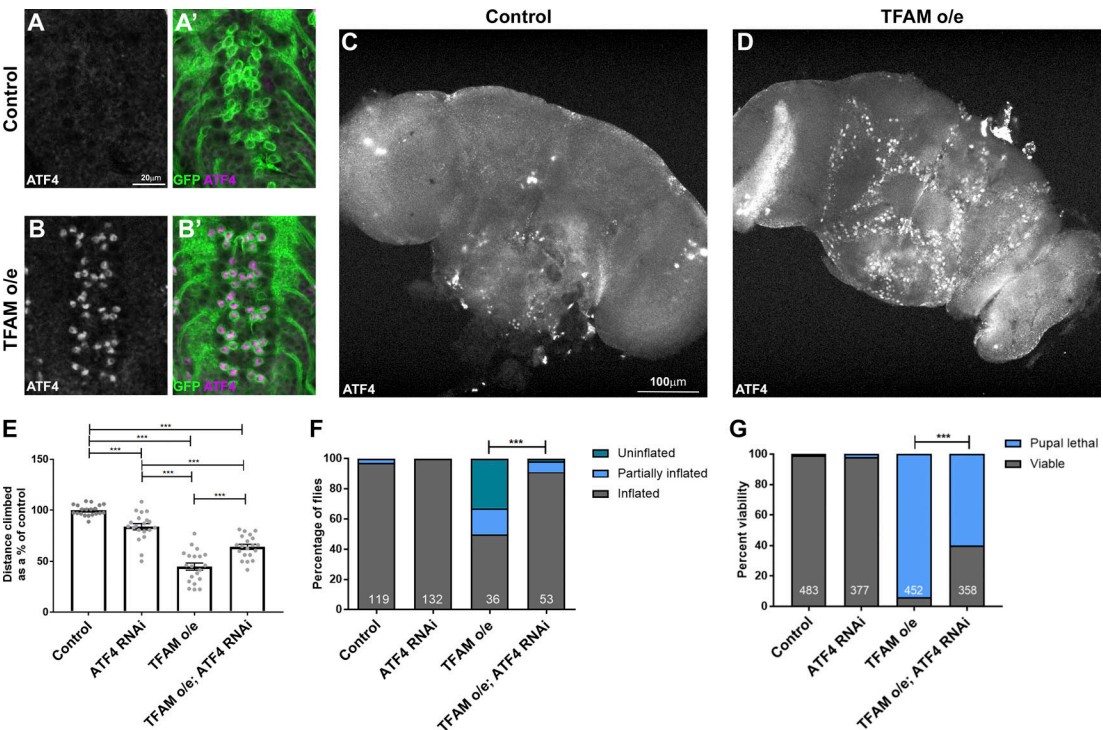

Figure 1. **Mitochondrial stress signaling via ATF4 in neurons. (A–B')** ATF4 is not expressed in control larval neurons (A and A'), but its expression is activated in motor neurons overexpressing (o/e) TFAM (B and B') using *OK371-Gal4*. ATF4 in white in A and B and magenta in A' and B'. CD8-GFP (green) marks motor neurons. **(C and D)** ATF4 is not expressed in the adult brain (C) but is activated in neurons overexpressing TFAM with *nSyb-Gal4* (D). **(E and F)** Knockdown of ATF4 alleviates the climbing (E) and wing inflation (F) defects caused by TFAM overexpression in motor neurons using *D42-Gal4*. Numbers of flies for wing inflation are in white. **(G)** Knockdown of ATF4 alleviates the pupal lethality caused by pan-neuronal TFAM overexpression with *nSyb-Gal4*. Number of pupal cases are in white. Controls are *Gal4* hemizygotes. Data in E are presented as mean ± SEM. *n* = 20 for all genotypes. ***, P < 0.001.

its role in the developing eye in *Drosophila*, ATF4 is expressed in photoreceptor neurons (Fig. S1 A; Kang et al., 2017), but is not normally expressed in the *Drosophila* larval or adult central nervous system (CNS; Fig. 1, A and C). Consistent with other mitochondrial stress models, TFAM overexpression in neurons causes activation of ATF4 expression in both the larval and adult CNS (Fig. 1, A–D; and Fig. S1, B and C).

TFAM overexpression also causes loss of presynaptic mitochondria, a reduction in presynaptic active zones, reduced climbing ability in adult flies, and inhibition of wing inflation, which is regulated by activity-dependent neuropeptide release from the CCAP neurons (Cagin et al., 2015; Duncan et al., 2018). To determine the functional role of ATF4, we performed climbing and wing inflation assays on flies overexpressing TFAM together with ATF4 knockdown in neurons. Expression of an RNAi targeting a control heterologous gene (luciferase) does not affect the TFAM overexpression climbing and wing inflation phenotypes (Fig. S1, D and E). However, ATF4 knockdown (Fig. S1 F) alleviates the climbing and wing inflation defects caused by neuronal TFAM overexpression (Fig. 1, E and F). Moreover, pan-neuronal overexpression of TFAM causes almost complete late pupal lethality, but knockdown of ATF4 in combination with TFAM overexpression significantly improves viability (Fig. 1 G). Therefore, ATF4 contributes to the reduced neuronal activity caused by mitochondrial stress.

**Mitochondrial stress reprograms neuronal metabolism**

We next characterized the gene expression changes caused by neuronal mitochondrial stress, and assessed the contribution of ATF4. Transcriptomic analysis of nervous system tissue showed that pan-neuronal TFAM overexpression significantly misregulates 87 genes with a range of different functions (Fig. 2 A and Table S1). Overexpression of ATF4 misregulates the expression of 149 genes (Fig. 2 B and Table S2). 24 of these ATF4 target genes are also misregulated by TFAM overexpression, and their expression is highly correlated (Fig. 2, B and C; and Table S3), consistent with activation of ATF4 by neuronal mitochondrial stress (Fig. 1, A–D). Importantly, knockdown of ATF4 reverses ~25% (22/87) of the transcriptional changes caused by TFAM overexpression (Fig. 2, A and D; and Tables S4 and S5; see Materials and methods), indicating that ATF4 is a key regulator of mitochondrial stress signaling in the *Drosophila* nervous system.

ATF4 regulates the expression of metabolic genes (Bao et al., 2016), including lactate dehydrogenase (LDH; *Impl3*) in *Drosophila* (Lee et al., 2015). Several of the genes whose expression is increased by TFAM overexpression but reversed by ATF4 knockdown are involved in glycolytic metabolism, including LDH (*Impl3*) and the trehalose transporter (*Tret1-1*; trehalose is the main circulating sugar in insects; Fig. 2 D and Table S5). To determine how mitochondrial stress affects neuronal metabolism, we performed metabolomic analysis using adult heads

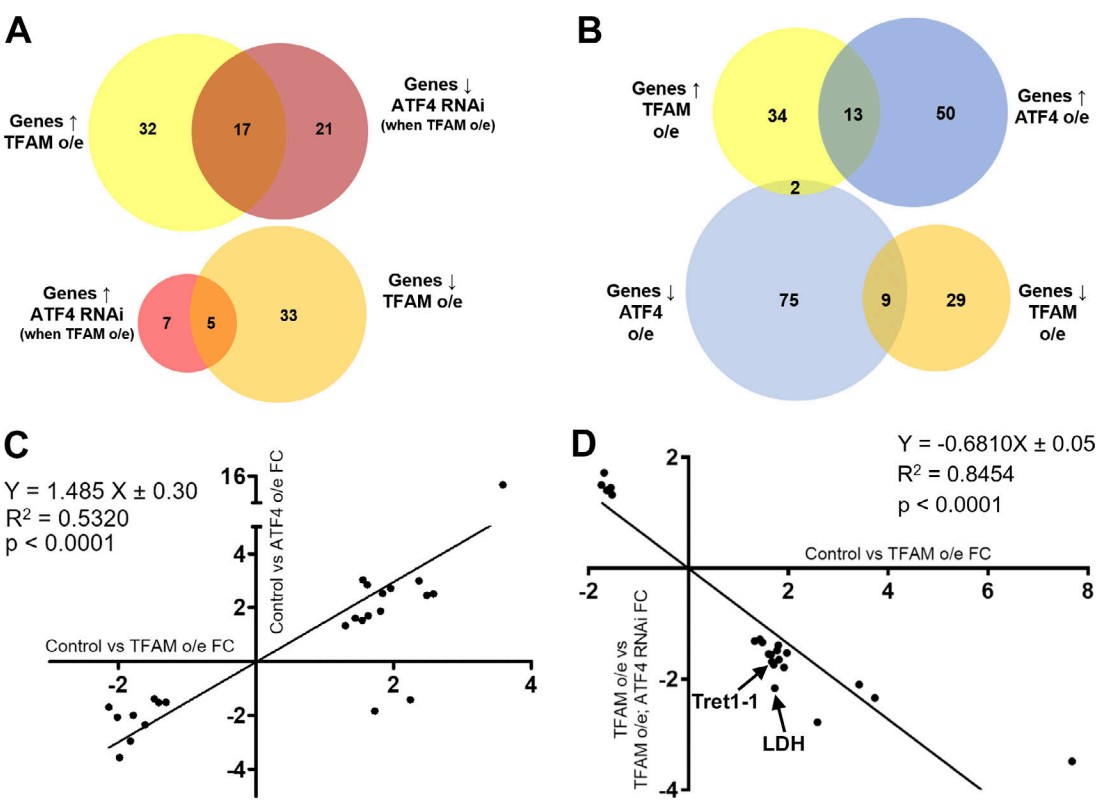

Figure 2. **Neuronal mitochondrial stress activates a transcriptional response via ATF4. (A)** Venn diagrams showing the number of significantly misregulated genes in the control versus TFAM overexpression (o/e) and TFAM overexpression versus TFAM overexpression combined with ATF4 RNAi conditions as determined by RNA sequencing. ↑ = up-regulated; ↓ = down-regulated. **(B)** Venn diagrams showing the number of significantly misregulated genes in the control versus TFAM overexpression and control versus ATF4 overexpression conditions. **(C)** Gene expression correlations between genes significantly misregulated in both the control versus TFAM overexpression and control versus ATF4 overexpression conditions. **(D)** Gene expression correlations between genes significantly misregulated in both the control versus TFAM overexpression and TFAM overexpression versus TFAM overexpression combined with ATF4 RNAi conditions. FC, fold change. See Materials and methods and Tables S1, S2, S3, S4, and S5 for details.

from flies with pan-neuronal TFAM overexpression. Pan-neuronal TFAM overexpression causes almost complete late pupal lethality (Fig. 1 G), so TFAM was pan-neuronally overexpressed in a $TFAM^{cO1716}$ loss-of-function heterozygous background, which produces viable adults. Analysis of global changes in metabolites shows that levels of 21 metabolites are altered by pan-neuronal TFAM overexpression (Fig. 3 A). Consistent with other mitochondrial stress models (Bao et al., 2016; Khan et al., 2017), serine and carnitine levels are increased in TFAM over-expressing tissue (Fig. 3 A). Grouping individual metabolites by metabolic pathway shows that levels of the TCA cycle intermediates α-ketoglutarate (α-KG) and fumarate are decreased (Fig. 3, B and C; and Table S6). The glycolytic pathway intermediates phosphoenolpyruvate, D-glyceraldehyde-3-phosphate, and the 3-phosphoglycerate precursor 2/3-phosphoglycerate are all decreased by TFAM overexpression, while lactate levels are increased (Fig. 3, D–F; and Table S6). These data provide evidence that mitochondrial stress causes defects in the TCA cycle and glycolytic metabolism in *Drosophila* neurons.

### Mitochondrial stress–induced 2-hydroxyglutarate (2-HG) is ATF4-dependent

Strikingly, the metabolite with the greatest fold change in TFAM overexpression tissue is 2-HG (Fig. 3 A and Table S6). L-2-HG is

the predominant enantiomer of 2-HG in *Drosophila* and is synthesized from α-KG by the promiscuous activity of LDH (Li et al., 2017). Since ATF4 knockdown reverses the mitochondrial stress–induced increase in LDH (*Impl3*) expression (Fig. 2 D and Table S5), we tested whether knockdown of ATF4 affects 2-HG levels. We find that knockdown of ATF4 significantly reduces the accumulation of 2-HG caused by TFAM overexpression (Fig. 3 G). Therefore, ATF4 regulates 2-HG levels in the nervous system during mitochondrial stress.

### L-2-HG regulates neuronal function

The ability of ATF4 knockdown to regulate 2-HG and alleviate the neuronal functional defects caused by TFAM overexpression suggested that 2-HG might contribute to neuronal dysfunction. LDH uses α-KG as a promiscuous substrate to produce L-2-HG, but L-2-HG is converted back to α-KG by the dedicated enzyme L-2-HG dehydrogenase (L-2-HGDH; Kranendijk et al., 2012; Li et al., 2017). A deficiency in L-2-HGDH in humans can lead to a disease known as L-2-hydroxyglutaric aciduria (L-2-HGA). L-2-HGA is a rare neurodegenerative disorder that causes hypotonia, tremors, epilepsy, delayed brain development, and psychomotor regression (Kranendijk et al., 2012). Chiral derivitization confirmed that the predominant 2-HG enantiomer present in TFAM-overexpressing tissue is L-2-HG (Fig. S2). Measurement

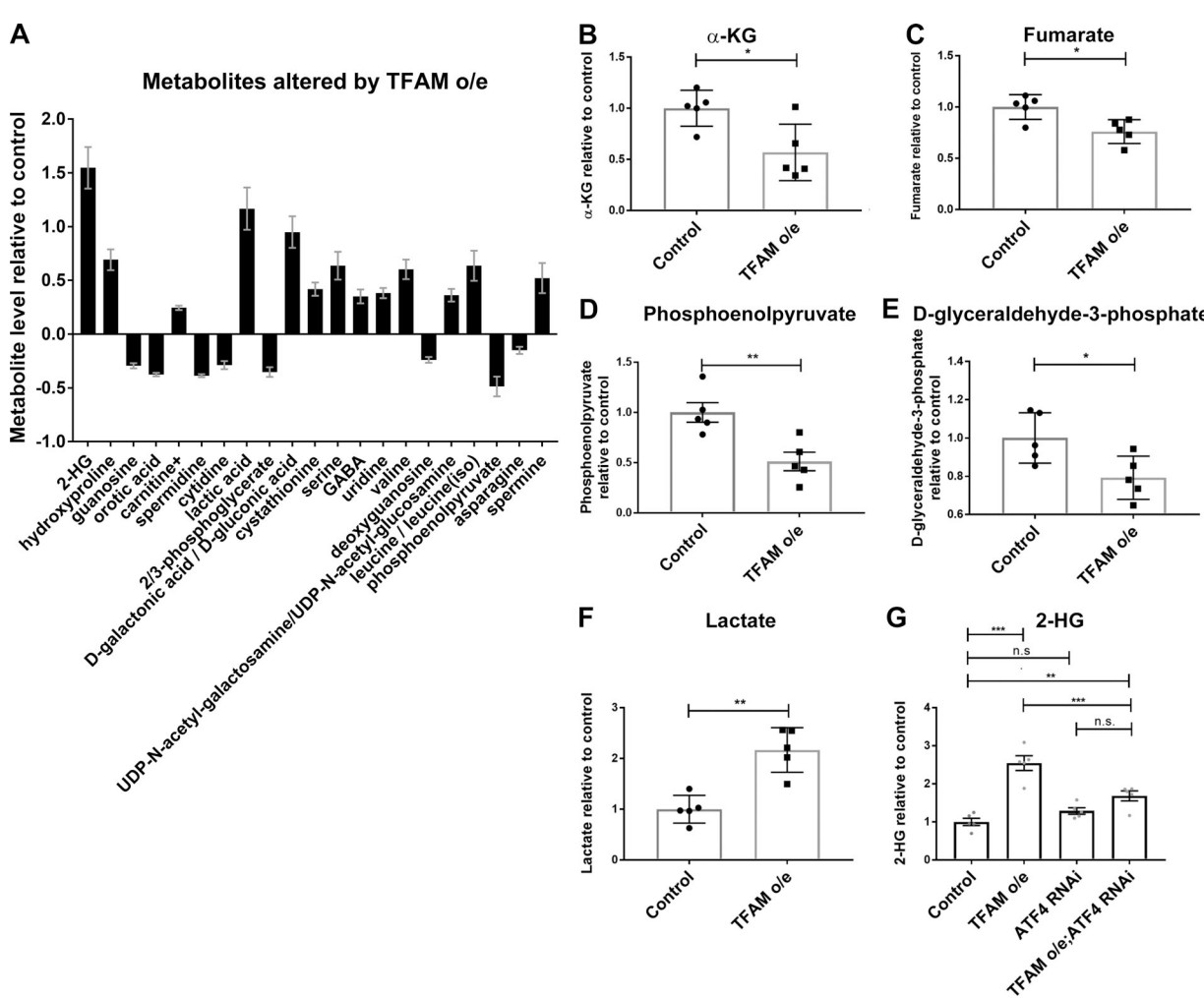

**Figure 3. Mitochondrial stress signaling increases 2-HG levels via ATF4. (A)** Metabolites whose levels are significantly changed (5% false discovery rate) in adult heads from flies with pan-neuronal TFAM overexpression (o/e) using *nSyb-Gal4*. **(B–F)** Levels of individual TCA cycle and glycolytic metabolites in adult heads from flies with pan-neuronal TFAM overexpression using *nSyb-Gal4*. See Table S6 for details. **(G)** Knockdown of ATF4 reduces the increase in 2-HG levels caused by pan-neuronal TFAM overexpression using *nSyb-Gal4*. Controls are *nSyb-Gal4* hemizygotes. Data are represented as mean ± SEM. *n* = 5 for all genotypes. ns, not significant; *, P ≤ 0.05; **, P ≤ 0.01; ***, P ≤ 0.001.

of 2-HG levels in fly heads overexpressing TFAM and with L-2-HGDH knockdown shows that the mitochondrial stress–induced increase in 2-HG is exacerbated by L-2-HGDH knockdown (Fig. 4 A). Functional assays show that knockdown of L-2-HGDH also exacerbates the climbing and wing inflation defects caused by TFAM overexpression (Fig. 4, B and C). Furthermore, overexpression of L-2-HGDH together with TFAM reduces 2-HG to control levels (Fig. 4 D) and completely rescues the climbing and wing inflation defects caused by TFAM overexpression (Fig. 4, E and F). Therefore, mitochondrial stress–induced activation of ATF4 and the consequent increase in 2-HG cause neurological dysfunction in *Drosophila*.

### Mitochondrial stress disrupts calcium homeostasis and activates ATF4 via the ER UPR

ATF4 is regulated at the level of translation by phosphorylation of eukaryotic initiation factor 2α (eIF2α). Overexpression of TFAM causes increased levels of phospho-eIF2α (Fig. 5, A–C). In *Drosophila*, eIF2α can be phosphorylated by the kinases PERK

and GCN2 (Malzer et al., 2013). The mitochondrial stress–induced activation of ATF4 is abolished by knockdown of PERK, but not GCN2 (Fig. 5, D–H; and Fig. S3, A–O). The PERK pathway is a branch of the ER UPR. To determine whether mitochondrial stress activates other branches of the ER UPR, we used a construct that reports on alternative splicing of the transcription factor XBP1 (Ryoo et al., 2007). XBP1 is a component of the IRE1 pathway and is alternatively spliced upon activation of the ER UPR. TFAM overexpression causes an increase in the alternative splicing of XBP1 (Fig. 5, I–K). Taken together, these data show that in *Drosophila* neurons, ATF4 activation, and the consequent increase in L-2-HG caused by mitochondrial stress are a result of activation of the ER UPR.

Increased ROS can activate mitochondrial stress response pathways, including the ER UPR (Quirós et al., 2016), but we previously showed that TFAM overexpression causes reduced glutathione redox potential in larval neurons (Cagin et al., 2015). Consistent with this finding, pan-neuronal TFAM overexpression causes decreased hydrogen peroxide levels in adult

Figure 4. **Mitochondrial stress–induced L-2-HG regulates neuronal function. (A)** The increase in 2-HG levels caused by pan-neuronal TFAM overexpression (o/e) is exacerbated by L-2-HGDH knockdown. *n* = 6 for all genotypes. **(B)** L-2-HGDH knockdown enhances the climbing defect caused by pan-neuronal TFAM overexpression. Control *n* = 13, L-2-HGDH RNAi *n* = 13, TFAM overexpression *n* = 12, TFAM overexpression + L-2-HGDH RNAi *n* = 10. **(C)** L-2-HGDH knockdown enhances the wing inflation defect caused by pan-neuronal TFAM overexpression. Numbers of flies are shown in white. **(D)** L-2-HGDH overexpression prevents the increase in 2-HG levels caused by TFAM overexpression. *n* = 4 for all genotypes. **(E)** L-2-HGDH overexpression alleviates the climbing defect caused by pan-neuronal TFAM overexpression. Control *n* = 14, L-2-HGDH overexpression *n* = 11, TFAM overexpression *n* = 12, TFAM overexpression + L-2-HGDH overexpression *n* = 14. **(F)** L-2-HGDH overexpression alleviates the wing inflation defect caused by pan-neuronal TFAM overexpression. Numbers of flies are shown in white. *nSyb-Gal4* was used for pan-neuronal expression. Controls are *nSyb-Gal4* hemizygotes. Data are represented as mean ± SEM. n.s., not significant; *, P ≤ 0.05; **, P ≤ 0.01; ***, P ≤ 0.001.

heads, which is not modified by ATF4 knockdown (Fig. S3 P), indicating that increased ROS do not mediate mitochondrial stress signaling via ATF4 in *Drosophila* neurons.

Insight into the mechanism by which mitochondrial stress activates the ER UPR may come from examination of the shared functions of mitochondria and the ER. Both organelles have critical roles in Ca²⁺ storage and buffering, and altered cellular Ca²⁺ homeostasis activates the ER UPR (Paillusson et al., 2016). To investigate whether mitochondrial stress disrupts cellular Ca²⁺ handling in *Drosophila* neurons, GCaMP6 was used to measure cytosolic Ca²⁺ in vivo. We find that TFAM overexpression does not alter basal cytosolic Ca²⁺ levels (Fig. S3, Q–S). However, GCaMP6 fluorescence is significantly increased in TFAM-overexpressing neurons during repetitive evoked neuronal activity (Fig. 5 L). This demonstrates that mitochondrial stress disrupts Ca²⁺ flux, and is therefore a potential mechanism by which the ER UPR and ATF4 are activated in *Drosophila* neurons.

Mitochondrial stress activates signaling pathways that enable mitochondria to control nuclear gene expression and cellular metabolism (Hunt and Bateman, 2018; Mehta et al., 2017). We have shown that in the *Drosophila* nervous system, mitochondrial stress disrupts Ca²⁺ homeostasis and triggers the ER UPR, resulting in activation of ATF4. ATF4 is a key element of the mitochondrial stress transcriptional response and reprograms neuronal metabolism. One result of the ATF4-dependent switch in metabolism is an increase in L-2-HG. Under conditions of mitochondrial stress, decreasing L-2-HG levels improves neuronal function, while increasing L-2-HG levels exacerbates neuronal dysfunction. Therefore, mitochondrial stress–induced L-2-HG regulates neuronal function in *Drosophila*.

The mitochondrial stress–induced increase in L-2-HG is conserved in other *Drosophila* models, mammalian cellular models, and patients with respiratory chain deficiency. Mutation of *Pink1*, which regulates mitochondrial quality control and is mutated in early onset familial Parkinson's disease, or the mitochondrial citrate carrier gene *scheggia* in *Drosophila*, cause increased 2-HG (Li et al., 2018; Tufi et al., 2014). 2-HG levels are also increased by loss of the complex III subunit Rieske iron-sulfur protein in mouse regulatory T cells, and L-2-HG is increased in cells from patients with a form of the mitochondrial disease Leigh syndrome (Ansó et al., 2017; Burr et al., 2016; Weinberg et al., 2019). Metabolomic analysis has also shown that patients with mutations in respiratory chain complex I, complex III, or multiple complex deficiencies have increased 2-HG levels (Reinecke et al., 2012).

L-2-HG inhibits α-KG–dependent oxygenases, such as the histone lysine demethylase JMJD2A and hypoxia-inducible factor prolyl hydroxylases, and so has wide-ranging effects on cell physiology (Burr et al., 2016; Chowdhury et al., 2011; Losman and Kaelin, 2013). In addition, L-2-HG can bind to and inhibit mitochondrial ATP synthase, indirectly inhibiting mechanistic target of rapamycin signaling (Fu et al., 2015). In *Caenorhabditis elegans*, the moderate inhibition of mitochondrial function and mechanistic target of rapamycin signaling caused by increased L-2-HG results in extended lifespan (Fu et al., 2015). However, mice that are mutant for *L-2-HGDH* have high levels of L-2-HG in the brain, white matter abnormalities, and late-onset neurodegeneration (Ma et al., 2017). Patients with L-2-HGA have progressive encephalopathy, particularly affecting the cerebellar white matter (Kranendijk et al., 2012). The effects of high levels of L-2-HG are likely to be context-dependent, but neurons seem

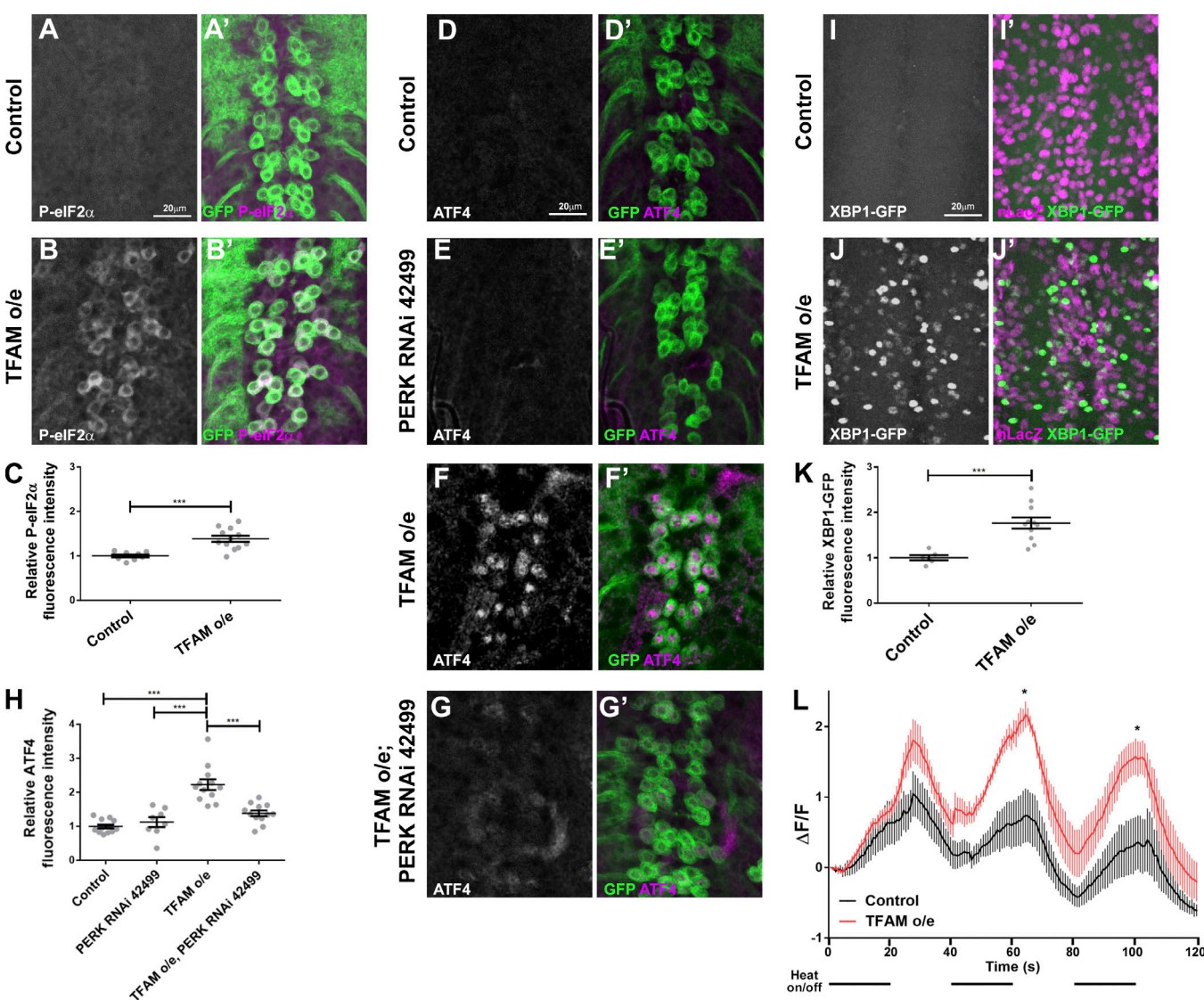

Figure 5. **Mitochondrial stress–induced activation of ATF4 via the ER UPR. (A–B')** Compared with controls (A and A'), overexpression of TFAM in larval motor neurons causes increased expression of phospho-eIF2α (P-eIF2α; B and B'). P-eIF2α in white in A and B and magenta in A' and B'. CD8-GFP (green) marks motor neurons. **(C)** Quantification of P-eIF2α expression. Control n = 8, TFAM overexpression n = 12. **(D–G')** Knockdown of PERK (RNAi line 42499) prevents the increase in ATF4 expression (white in D–G and magenta in D'–G') caused by overexpression (o/e) of TFAM in larval motor neurons using *OK371-Gal4*. CD8-GFP expression (green) marks motor neurons. **(H)** Quantification of ATF4 expression. Control n = 12, PERK RNAi n = 8, TFAM overexpression n = 12, TFAM overexpression + PERK RNAi n = 12. **(I–J')** Compared with controls (I and I'), overexpression of TFAM in larval motor neurons using *OK371-Gal4* causes increased expression of XBP1-GFP (J and J'). XBP1-GFP in white in I and J and green in I' and J'. nLacZ (magenta) marks motor neurons. **(K)** Quantification of XBP1-GFP expression. Control n = 6, TFAM overexpression n = 11. **(L)** TFAM overexpression in larval motor neurons disrupts activity-dependent Ca²⁺ buffering. Mean of traces for GCaMP6m fluorescence in motor neuron cell bodies (using *OK371-Gal4*) as TrpA1 channels were activated by 20-s on/off pulses of 25°C heat, as indicated by the horizontal black lines below the graph (line = heat on, space = heat off). Control = black trace (n = 5), TFAM overexpression = red trace (n = 5). Controls are *Gal4* hemizygotes. Data are represented as mean ± SEM. *, P ≤ 0.05; ***, P ≤ 0.001.

particularly sensitive to this metabolite. Our functional analyses suggest that modulating L-2-HG levels may be a potential therapeutic strategy for neurological diseases associated with mitochondrial dysfunction or persistent ATF4 activation.

## Materials and methods
### Fly strains and growth conditions
Flies were maintained on standard yeast, glucose, cornmeal, and agar food at 25°C in a 12-h light/dark cycle unless stated

otherwise. Details of fly stocks are listed in Table S7. Fly stocks were from the Bloomington Drosophila Stock Center; the Vienna Drosophila Resource Center (Dietzl et al., 2007); the NIG-Fly Stock Center, Japan; and FlyORF. Pan-neuronal TFAM overexpression causes late pupal lethality, so to obtain viable adults, TFAM was pan-neuronally overexpressed in a *TFAM* loss-of-function heterozygous mutant background (*nSyb-Gal4, TFAM^{c01716}*). For imaging experiments, embryos were laid over a 24-h period at 25°C, incubated for a further 24 h at 25°C, and then incubated at 29°C for 3 d before analysis, unless stated otherwise.

## Behavioral analysis

For all assays using flies with neuronal dysfunction, vials were placed on their sides during eclosion to prevent flies from becoming stuck in the food. For climbing assays, male flies of appropriate genotype were selected under $CO_2$ anesthesia, within 24 h of eclosion. Assays were undertaken 24–48 h later to allow the flies to recover from the anesthetic, 1–3 h after morning illumination. Individual flies were aspirated from a vial into a 10-ml serological pipette (Falcon). Flies were relocated to the base of the pipette by tapping against the bench. The height obtained in three continuous 10-s climbs was measured for each fly, and the mean computed to give the data point for that insect. Wandering (i.e., nonvertical) or discontinuous climbs were excluded.

For wing inflation assays, all flies that eclosed from each cross were collected under $CO_2$ anesthesia, and stored at 25°C for ≥24 h to ensure that an uninflated wing phenotype was not due to young age. The numbers of flies with inflated, semi-inflated, and uninflated wings were then counted.

Viability assays were performed by counting the number of pupae containing flies that failed to eclose and empty pupal cases from which viable flies has successfully eclosed.

## Immunofluorescence and imaging

Images were taken using a Zeiss LSM710 confocal microscope with Zen 2012 LSM software or a Nikon A1R confocal microscope with NIS Elements software, as described below. Images were taken at room temperature unless stated otherwise. Imaging of controls and experimental samples in each experiment was performed using identical confocal microscope settings.

For larval CNS imaging, fly crosses were left to egg-lay at 25°C for 3 d. The adults were then removed and the embryos incubated at 29°C (to enhance Gal4 activity) until dissection at the third instar larval stage. For fixed CNS tissue imaging, larvae were removed from their vials, cleaned in ice-cold PBS (Oxoid; Thermo Fisher Scientific), and then transferred to a drop of PBS on a Sylgard plate. The larvae were bisected with forceps (Agar Scientific) and the mouth end retained in the PBS. The cuticle was then inverted by coaxing the open end of the larvae over forceps holding the mouth parts. After the viscera was removed, the inverted cuticle with attached CNS was transferred to a 0.5-ml Eppendorf and fixed for 25 min in 4% formaldehyde (Thermo Fisher Scientific)/PBS. After three 10-min washes with 0.1% Triton X-100 (Sigma-Aldrich)/PBS (PBS-T), the samples were blocked for 1 h in 5% normal goat serum (Yorlab)/PBS-T. The relevant primary antibodies were applied in 5% normal goat serum/PBS-T and left rocking overnight at 4°C. After three further 10-min PBS-T washes, the samples were incubated with secondary antibody/PBS-T at room temperature for 1 h, and then washed 2× with PBS-T and 1× with PBS (10 min per wash). The samples were then transferred in PBS onto SuperFrost Plus microscope slides (Thermo Fisher Scientific), where the cuticle was removed and the CNS mounted dorsal side up in Vectashield (Vector Laboratories) under 22 × 22-mm size-0 coverslips (Academy). The slides were stored at 4°C in the dark. Images were taken on a Zeiss LSM710 confocal microscope with an EC Plan Neofluar 40× NA 1.3 oil immersion lens.

For imaging adult brains, crosses were incubated at 25°C. 2–5 d after eclosion, each adult fly was briefly anaesthetized using $CO_2$, then immobilized ventral side up with a micro-pin (Entomoravia) through the thorax on Sylgard in PBS. Adult brains were then dissected using fine tweezers, fixed for 30 min in 4% formaldehyde (Thermo Fisher Scientific)/PBS, and stained in the same way as larval CNS tissue. Images were taken using a Nikon A1R confocal microscope with a CFI PLAN APO VC 20× NA 0.75 lens.

Primary antibodies were P-eIF2α (1:500; rabbit anti-phospho-eIF2α [Ser51]; Cell Signaling Technology 9721), ATF4 (1:250; rat; this study), β-gal (1:1,000; mouse anti-β-galactosidase; Z378A; Promega). Secondary antibodies were goat anti-mouse Alexa Fluor 633 (A21052; Invitrogen), goat anti-rabbit Alexa Fluor 546 (A11035; Invitrogen), goat anti-rat Alexa Fluor 555 (A21434; Thermo Fisher Scientific), and Cy5 AffiniPure donkey anti-rat (JIR 712-175-153).

For quantification of fixed immunostained tissue with membrane marker (CD8-GFP), maximum intensity projections of z-stacks through the dorsal side of the ventral nerve cord were obtained in ImageJ. 50 points in the nuclear or cytoplasmic region of interest were then selected in the marker channel (CD8-GFP) without reference to the channel of interest. The average fluorescence intensity of those 50 points was then obtained in the channel of interest.

For basal GCaMP6m imaging, larvae were dissected in Schneider's medium (Sigma-Aldrich) and the CNS then transferred to a SuperFrost Plus microscope slide (Thermo Fisher Scientific). A coverslip was applied with 15 µl of Schneider's underneath to hydrate the sample and sealed with nail polish. z-Stacks were taken with a Zeiss LSM710 confocal microscope, a 20×, NA 0.8 Plan Apochromat objective, using the 488 nm laser, with 1-µm inter-slice intervals. Maximum intensity projections were created in ImageJ. 50 points were selected in fluorescent regions, and the average of those points was computed to give the mean fluorescence for each sample.

For thermogenetically stimulated GCaMP6m imaging, wandering third instar larvae were bisected and the cuticle inverted in hemolymph-like 3.1 (HL3.1) media (70 mM NaCl, 5 mM KCl [Thermo Fisher Scientific], 1.5 mM $CaCl_2$ [Sigma-Aldrich], 4 mM $MgCl_2$ [Sigma-Aldrich], 10 mM $NaHCO_3$ [Life Technologies], 5 mM Trehalose [Thermo Fisher Scientific], 115 mM sucrose [Sigma-Aldrich], and 5 mM Hepes [Alfa Aesar]). The CNS was dissected from the inverted cuticle and retained on the microscope slide in residual HL3.1. A 1-mm-thick, 10-mm–external diameter rubber O-ring was placed around the CNS to serve as a mold for the agarose that was used to immobilize the sample during imaging. The O-ring was flooded with molten 1% low-melt agarose (2-hydroxyethyl agarose [Sigma-Aldrich] in HL3.1 media) ejected from a P1000 pipette with sufficient force to carry the CNS to the surface of the agarose. The slide was then placed in the refrigerator for ~5 min to allow the agarose to set, after which the O-ring was carefully removed and a 22 × 22-mm size-0 coverslip was placed directly onto the agarose. Activation of the TrpA1 channel was achieved via a peltier (Hebei TEC1-12706) secured flush underneath the microscope slide during imaging. The peltier was connected to the Thurlby Thandar

PL154 powerpack (RS Components) and calibrated to ramp to a temperature of 25°C using an infrared thermometer (GM320; L-FENG-UK) pointed at the surface of the peltier while the voltage was adjusted. Time series were taken with a Zeiss LSM710 confocal microscope at 2 Hz using a 20×, NA 0.8, Plan Apochromat objective and a 488-nm laser. Times series were quantified by plotting the profile of 50 points (selected within the cell bodies) across each frame in ImageJ.

## Amplex red assay

10 Snap-frozen adult fly heads were ground using a plastic pestle in 300 µl of Amplex red buffer (50 µM Amplex red [Thermo Fisher Scientific] and 1 U/ml HRP [Sigma-Aldrich] in PBS). Samples were incubated in the dark at 37°C for 1 h, with mixing every 15 min. Samples were then centrifuged at 14,000 rpm for 1 min, and 200 µl supernatant per well was added to a 96-well plate. Absorbance was read at 560 nm using a CLARIOstar microplate reader (BMG Labtech). For each genotype, five biological replicates were analyzed and the average absorbance was calculated.

## ATF4 antibody creation

A 180-bp region common to all isoforms of *Drosophila* ATF4, but upstream of the conserved DNA binding domain, was amplified from genomic DNA using primers 5′-ATATT<u>GGATCC</u>GAATGT CTTTTTGGACCAAAAGGC-3′ and 5′-AGCGA<u>CTCGAG</u>AGCCAT CATTGAGCTGGTAAT-3′, digested with BamHI and XhoI (enzyme sites underlined in primers) and cloned into pGEX4T-2 (Amersham). The ATF4-GST fusion protein was overexpressed in BL21 (DE3) cells and purified using glutathione sepharose beads (4B; GE Healthcare). Purified ATF4-GST was used to immunize Sprague Dawley rats (Envigo RMS UK) and generate antisera.

## Quantitative RT-PCR (qRT-PCR)

qRT-PCR was performed as described previously (Cagin et al., 2015) but using the following primers: ATF4 forward, 5′-ATT CACTGCTGCCGCAAAA-3′; ATF4 reverse, 5′-GTTCAACGTTGC CTTTTGGT-3′; Rpl4 forward, 5′-TCCACCTTGAAGAAGGGCTA-3′; and Rpl4 reverse, 5′-TTGCGGATCTCCTCAGACTT-3′.

## RNA sequencing transcriptomic analysis

For each replicate, 20 CNSs were dissected from third instar larvae in cold PBS and placed into 100 µl of lysis buffer + β-mercaptoethanol from the Absolutely RNA Microprep kit (Agilent Technologies). Each genotype was prepared in quadruplicate. RNA was extracted from the CNS tissue using the Absolutely RNA Microprep kit according to the manufacturer's protocol. The samples were sent on dry ice to the Next-Generation Sequencing Laboratory at Glasgow Polyomics, where they prepared polyA cDNA libraries using the TruSeq Stranded mRNA kit (Illumina), and oversaw 75-bp single-end sequencing at a depth of 25 million reads on the Illumina NextSeq 500. The raw unaligned reads were received in fastq format and mapped to the *Drosophila melanogaster* dm6 genome using Bowtie2 (Galaxy Version 2.3.4.2) software (Trapnell et al., 2012). Alignment quality was ≥92% overall read alignment for all samples. Raw fragment counts were generated from the BAM files via the featureCounts (Galaxy Version 1.6.3+galaxy2) program (Liao et al., 2014) and fed into the DESeq2 program (Galaxy Version 2.11.40.6; Love et al., 2014) to analyze differential gene expression. Bowtie2, featureCounts, and DESeq2 were all accessed via the web-based user interface software Galaxy (http:// usegalaxy.org; Afgan et al., 2018) using default settings. RNA sequencing data are deposited in the GEO repository under accession no. GSE128244.

A gene was considered to be significantly up- or down-regulated if the fold change was greater than or equal to ±1.2, the adjusted P value <0.05 and the fragments per kilobase of transcript per million mapped reads ≥1 in at least one condition. The expression of a gene that was significantly up- or down-regulated by TFAM overexpression compared with control (Table S1) was considered to have been reversed by ATF4 knockdown (Table S5) if it showed significant, but opposite, regulation when the TFAM overexpression condition was compared with the TFAM overexpression combined with the ATF4 knockdown condition (Table S4).

## Metabolomic analysis

20 2–5-d-old adult flies (equal numbers of males and females) were snap-frozen on liquid nitrogen in a 15-ml Falcon tube and then vortexed for 5 s five times to decapitate. Heads were then quickly separated and stored at –80°C. Soluble metabolites were extracted directly from tissue using cold methanol/water (80/ 20, vol/vol) at 1 µl per 10 µg of tissue. Tissue was disrupted for 15 s by ultrasonication (Branson Sonifier 250). Debris were pelleted by centrifugation at 18,000 × $g$ for 15 min at 4°C. The supernatant was transferred to a new tube and evaporated to dryness using a SpeedVac concentrator (Thermo Fisher Scientific). Metabolites were reconstituted in 50% acetonitrile in analytical-grade water, vortex-mixed, and centrifuged to remove debris. Samples were analyzed by HPLC and high-resolution mass spectrometry and tandem mass spectrometry (MS/MS). Specifically, the system consisted of a Q-Exactive in line with an electrospray source and an Ultimate3000 (Thermo Fisher Scientific) series HPLC consisting of a binary pump, degasser, and auto-sampler outfitted with an Xbridge Amide column (dimensions of 4.6 × 100 mm and a 3.5-µm particle size; Waters). Mobile phase A contained 95% (vol/vol) water, 5% (vol/ vol) acetonitrile, 10 mM ammonium hydroxide, and 10 mM ammonium acetate, pH 9.0; and mobile phase B was 100% acetonitrile. The gradient was as follows: 0 min, 15% A; 2.5 min, 30% A; 7 min, 43% A; 16 min, 62% A; 16.1–18 min, 75% A; 18–25 min, and 15% A with a flow rate of 400 µl/min. The capillary of the electrospray ionization (ESI) source was set to 275°C, with sheath gas at 45 arbitrary units, auxiliary gas at 5 arbitrary units, and the spray voltage at 4.0 kV. In positive/negative polarity switching mode, a mass-to-charge ratio scan range from 70 to 850 was chosen, and MS1 data were collected at a resolution of 70,000. The automatic gain control target was set at $10^6$, and the maximum injection time was 200 ms. The top five precursor ions were subsequently fragmented in a data-dependent manner, using the higher energy collisional dissociation cell set to 30% normalized collision energy in MS2 at a

resolution power of 17,500. Resolution of enantiomers of 2-HG was accomplished by derivatization with diacetyl-L-tartaric anhydride (DATAN) in acetic acid after the above SpeedVac step and measured by HPLC-MS/MS as previously described (Oldham and Loscalzo, 2016).

For separation of DATAN derivitized D- and L-2-HG, samples were analyzed by HPLC-MS/MS, which consisted of a TSQ in line with an electrospray source and an Vanquish (Thermo Fisher Scientific) series HPLC consisting of a binary pump, degasser, and auto-sampler outfitted with an Ascentis Express C18 colume (dimensions of 2.1 × 150 mm and a 2.7-µm particle size; Supelco). Liquid chromatography was performed using a 98% buffer A (water with 2 mM amonium formate, pH 3.5, adjusted by formic acid) and 2% buffer B (methanol) isocratic elution of 10 min per sample. In negative mode, the capillary of the ESI source was set to 325°C, with sheath gas at 50 arbitrary units, auxiliary gas at 10 arbitrary units, and the spray voltage at 2,500 V. S-lens values were 37 V and reactions monitored, and collision energies were mass-to-charge ratio $363.06 \rightarrow 147.03$ (20 V).

Data acquisition and analysis were performed by Xcalibur 4.1 software and Tracefinder 4.1 software, respectively (both from Thermo Fisher Scientific). The peak area for each detected metabolite was normalized by the total ion count. 2-HG chromatograms were generated with Xcalibur software with spectral smoothing algorithm boxcar 15.

### Statistical analyses

GraphPad Prism (GraphPad Software) was used to create graphs and for statistical analysis. Pairwise comparisons were analyzed via unpaired two-tailed $t$ test, with Welch's correction applied if the variances of the samples were significantly different. Data requiring multiple comparisons were analyzed by parametric one-way ANOVA with Tukey's post hoc test. Global metabolite analysis was performed using the two-stage linear step-up procedure of Benjamini, Krieger, and Yekutieli (Benjamini et al., 2006), with a false discovery rate (Q) = 5%. $\chi^2$ tests were used to analyze categorical data. P values <0.05 were considered significant.

### Online supplemental material

Fig. S1 shows ATF4 expression, control climbing and wing inflation assays, and ATF4 qRT-PCR. Fig. S2 shows that 2-HG measured in heads from flies with pan-neuronal TFAM overexpression is predominantly the L enantiomer. Fig. S3 shows that mitochondrial stress–induced ATF4 activation requires PERK but not GCN2. Table S1 shows genes misregulated by TFAM overexpression. Table S2 shows genes misregulated by ATF4 overexpression. Table S3 shows genes that are commonly regulated by TFAM overexpression and ATF4 overexpression. Table S4 shows genes whose expression is modified by ATF4 knockdown. Table S5 shows genes whose expression is reversed by ATF4 knockdown. Table S6 shows metabolomic analysis of adult heads from flies with pan-neuronal (nSyb-Gal4) TFAM overexpression, ATF4 knockdown, or TFAM overexpression together with ATF4 knockdown and controls. Table S7 shows details of *Drosophila* lines.

## Acknowledgments

Stocks obtained from the Bloomington Drosophila Stock Center (National Institutes of Health P40OD018537), the Vienna Drosophila Resource Center, and FlyORF were used in this study. We are grateful to Jason Tennessen (Indiana University Bloominton, Bloomington, IN), Darren Williams (King's College London, London, UK), and Rita Sousa-Nunes (King's College London, London, UK) for fly stocks. We thank Peng Gao and the Robert H. Lurie Cancer Center Metabolomics Core at Northwestern University Feinberg School of Medicine for metabolomics analysis. We thank the Wohl Cellular Imaging Centre at King's College London for help with light microscopy.

This work was funded by Alzheimer's Research UK (ARUK-IRG2017A-2) to J.M. Bateman; L. Granat is supported by the UK Medical Research Council (MR/N013700/1) and King's College London Medical Research Council Doctoral Training Partnership in Biomedical Sciences; R.J. Hunt was funded by a PhD studentship from the Guy's and St Thomas' Charity. N.S. Chandel and G.S. McElroy were supported by National Institutes of Health grants AG049665-04 and T32CA9560-32, respectively. According to the UK research councils' Common Principles on Data Policy, all data supporting this study will be openly available at https://doi.org/10.1083/jcb.201904148.

The authors declare no competing financial interests.

Author contributions: Conceptualization: N.S. Chandel and J.M. Bateman; Methodology: R.J. Hunt and G.S. McElroy; Investigation: R.J. Hunt, L. Granat, G.S. McElroy, R. Ranganathan, and J.M. Bateman; Writing – original draft preparation: J.M. Bateman; Writing – review and editing: N.S. Chandel and J.M. Bateman; Visualization: R.J. Hunt, L. Granat, G.S. McElroy, R. Ranganathan, and J.M. Bateman; Supervision: N.S. Chandel and J.M. Bateman; Funding acquisition: N.S. Chandel and J.M. Bateman.

Submitted: 25 April 2019

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
