## [Reviewer comments · The Journal of Cell Biology]

Mitochondrial stress causes neuronal dysfunction via an ATF4-dependent increase in L-2-hydroxyglutarate

Rachel Hunt, Lucy Granat, Gregory McElroy, Ramya Ranganathan, Navdeep Chandel, and Joseph Bateman

Corresponding Author(s): Joseph Bateman, King's College London and Navdeep Chandel,

Review Timeline:

Submission Date:	2019-04-25
Editorial Decision:	2019-05-30
Revision Received:	2019-08-19
Editorial Decision:	2019-09-04
Revision Received:	2019-09-10

Monitoring Editor: Johan Auwerx

Scientific Editor: Melina Casadio

Transaction Report:

DOI: <https://doi.org/10.1083/jcb.201904148>

May 30, 2019

Re: JCB manuscript #201904148

Dr. Joseph Bateman
King's College London
125 Coldharbour lane
London SE5 9NU
United Kingdom

Dear Dr. Bateman,

Thank you for submitting your manuscript entitled "Mitochondrial stress-induced L-2-hydroxyglutarate regulates neuronal function". The manuscript was assessed by three expert reviewers, whose comments are appended to this letter. We invite you to submit a revision if you can address the reviewers' key concerns, as outlined here.

You will see that all of the reviewers are positive about the work. Reviewer 1 has relatively minor comments that should be addressed. Reviewer 2 and 3 both include suggestions for revisions. Please focus on those revisions that are aimed at supporting the existing conclusions as opposed to extending the work. Reports in JCB are intended for manuscripts offering novel findings with more limited mechanistic insights.

GENERAL GUIDELINES:

Text limits: Character count for a Report is < 20,000, not including spaces. Count includes title page, abstract, introduction, results, discussion, acknowledgments, and figure legends. Count does not include materials and methods, references, tables, or supplemental legends.

Figures: Reports may have up to 5 main text figures. To avoid delays in production, figures must be prepared according to the policies outlined in our Instructions to Authors, under Data Presentation, <http://jcb.rupress.org/site/misc/ifora.xhtml>. All figures in accepted manuscripts will be screened prior to publication.

Supplemental information: There are strict limits on the allowable amount of supplemental data. Reports may have up to 3 supplemental figures. Up to 10 supplemental videos or flash animations are allowed. A summary of all supplemental material should appear at the end of the Materials and methods section.

Our typical timeframe for revisions is three months; if submitted within this timeframe, novelty will not be reassessed at the final decision. Please note that papers are generally considered through only one revision cycle, so any revised manuscript will likely be either accepted or rejected.

Thank you for this interesting contribution to Journal of Cell Biology. You can contact us at the journal office with any questions, cellbio@rockefeller.edu or call (212) 327-8588.

Sincerely,

Johan Auwerx
Monitoring Editor
JCB

Rebecca Alvania
Executive Editor
JCB

Reviewer #1 (Comments to the Authors (Required)):

In this paper, Hunt et al. examine the metabolic mechanisms that lead to neuronal dysfunction in response to mitochondrial stress. The authors use an established model for inducing mitochondrial stress in neurons by overexpressing TFAM. They show that this results in up-regulation of the ATF4 transcription factor and that disruption of ATF4 by RNAi is sufficient to rescue the climbing defects and wing inflation phenotypes caused by TFAM overexpression. Using transcriptional profiling, the authors show that the elevated expression of many genes caused by TFAM overexpression is reversed by ATF4 RNAi. Interestingly, these genes include LDH, providing a possible metabolic link between ATF4 rescue and LDH activity. Consistent with this, the authors show that a major product from LDH in *Drosophila*, 2HG, increases upon TFAM overexpression and this change can be suppressed by ATF4 RNAi. In addition, changes in the expression of the enzyme that inactivates 2HG, L-2-HGDH, can modulate 2HG levels as well as climbing ability and wing inflation caused by TFAM overexpression. These results indicate that mitochondrial stress-induced induction of ATF4 causes neuronal dysfunction through changes in 2HG levels. The authors also tie this pathway to disrupted calcium flux as a possible mechanism that links mitochondrial stress to the unfolded protein response and ATF4 expression. This is a well written paper that is supported by clear and convincing data. I have only a few relatively minor suggestions to offer.

1. In Fig. 1F the authors depict genes that change expression upon TFAM overexpression but which are "reversed" by ATF4 RNAi. How do the authors define this? There is no description in the text of the criteria used to select the genes shown in the figures and supplemental tables.

2. Why do the authors shift between a motor neuron driver in Fig. 1 (D42-Gal4) and a pan-neuronal driver (nSyb-Gal4) for the rest of the studies?

3. Figure S2 and its accompanying legend would both benefit from revision and annotation to make them easier to interpret.

4. A minor sentence error: on pg. 5, "The mitochondrial stress-induced activation of ATF4 is abolished by knock-down [of] PERK, but not GCN2..."

Reviewer #2 (Comments to the Authors (Required)):

In this manuscript, the authors have investigated how mitochondrial stress can contribute to neuronal dysfunction. In particular, the authors demonstrated that in response to mitochondrial stress and the endoplasmic reticulum unfolded protein response, the transcription factor ATF4 is induced in the brain of *Drosophila*. ATF4 was shown to be partially responsible for reduced neuronal activity in flies through the accumulation of L-2-hydroxyglutarate (L-2-HG). Reducing L-2-HG levels by overexpressing L-2-HG dehydrogenase in the brain of flies with mitochondrial stress improved neurological function. While this study has some important implications for a better understanding of neurological diseases associated with mitochondrial dysfunction, the findings would be strengthened by further understanding of the mechanisms involved.

Specific comments:

a) The authors postulate that L-2-HG accumulates in response to ATF4 induction due to the promiscuous activity of LDH in neurons. In the literature, additional metabolic enzymes (i.e. MDH, PHGDH) have been reported to have a similar promiscuous activity. Are 3PG and malate levels altered upon ATF4 induction?

b) What are the effects of LDH loss on 2HG production?

c) How ATF4 regulates LDH-dependent 2HG production remains ill-defined. Is LDH expression altered upon ATF4 activation? Or does ATF4 induction promote cellular acidity or changes in redox?

d) Since an ATF4 antibody was generated, the ATF4 knockdown in Figure S1C should be validated by western blotting in addition to the shown mRNA levels.

e) The manuscript would be strengthened by further evidence that the *in vivo* phenotypes are connected to ATF.

Reviewer #3 (Comments to the Authors (Required)):

The manuscript by Hunt et. al. describes how mitochondria stress induced by overexpression of TFAM upregulates 2-hydroxyglutarate to regulate neuronal function. They further show that triggering mitochondrial stress in neurons leads to the activation of ATF4 which increases the level of 2-hydroxyglutarate in the brain. Overexpression of 2-hydroxyglutarate dehydrogenase decreases 2-hydroxyglutarate levels, and improves neuronal function. Perhaps the most exciting finding of this study is the observation that enhanced production of 2-hydroxyglutarate due to mitochondria dysfunction can cause neurodegeneration. Thus unlike many previous studies linking mitochondria dysfunction to neurodegeneration via increased ROS production, this report is of "particular novelty and high general interest" because it introduces a new metabolite as another

cause for neurodegeneration triggered as a result of mitochondrial dysfunction. Altogether, the work described in this manuscript is exciting and novel, although the link to Calcium homeostasis discussed towards the end is a bit tenuous. However, given that this was submitted as a report and not a full article, I will recommend that this be considered for publication if the following concerns are resolved:

1. In figure 1, the authors overexpressed TFAM in larval motor neurons using OK371-Gal4 (1A-C), and beautifully showed that this results in nuclear accumulation of ATF4. However, in 1D-E, they performed their locomotory assays in adult flies using a different Gal4 driver (D42-Gal4). Are there any locomotory assays the authors can perform on larvae to directly correlate the larval TFAM-dependent nuclear localization of ATF4 with a locomotory defect? Alternatively, nuclear localization of ATF4 should be shown in neurons overexpressing TFAM using the D42-Gal4 line. By using the same Gal4 line to show both phenotypes described, it will help directly correlate the locomotory defect with TFAM/ATF4 overexpression.
2. Many of the figures involved pan-neuronal expression of transgenes using nSyb-Gal4. This seems reasonable as the paradigm used by the authors allowed the recovery of viable adults. The authors should repeat the experiments in Figure 4A-K with nSyb-Gal4, to rule out possible developmental defects giving rise to some of the phenotypes described in Figure 4. For instance, the increase in phospho-eIF2a in response to TFAM overexpression should be shown as a western blot on head extracts overexpressing TFAM with nSyb-Gal4, etc.
3. Similarly, use western blot on head extracts to resolve whether nSyb-Gal4 mediated overexpression of TFAM and PERK RNAi suppress the enhanced ATF4 expression evident when TFAM alone is overexpressed. Along the same lines can the enhanced GFP signal of XBP1-GFP be reproduced on western blots from head extracts?
4. The authors explored the possibility that alterations in Calcium signaling could trigger some of the phenotypes observed. However, given the plethora of literature connecting redox signaling to neurodegeneration, ROS levels should also be quantified in the head extracts by the Amplex red assay to ascertain if ROS levels are affected as well. Its possible that 2-HG induces oxidative stress to cause neurodegeneration.
5. Is the increase in 2-hydroxyglutarate observed when TFAM is overexpressed a general feature of mitochondrial stress? Do other paradigms of mitochondria stress such as expression of RNAi to OXPHOS proteins or mitochondrial ribosomal proteins cause an increase in 2-hydroxyglutarate levels?

We are very grateful to the Editor and reviewers for their helpful comments. We have addressed all the reviewers' comments by the addition of new phenotypic, imaging and transcriptomic data, as well as additional discussion. A detailed list of our responses to the reviewers' comments and description of the changes made to the manuscript are given below. We have also highlighted the changes in the text of the manuscript.

Reviewer #1

1. In Fig. 1F the authors depict genes that change expression upon TFAM overexpression but which are "reversed" by ATF4 RNAi. How do the authors define this? There is no description in the text of the criteria used to select the genes shown in the figures and supplemental tables.

-The Venn diagram in Figure 1F (now Figure 2A) shows the numbers of genes whose expression is significantly increased or decreased in the control versus TFAM overexpression condition and the TFAM overexpression versus TFAM overexpression combined with ATF4 RNAi condition. We define genes that change expression upon TFAM overexpression but which are "reversed" by ATF4 RNAi as genes that are significantly mis-regulated by TFAM overexpression compared to control and also significantly, but oppositely, regulated when the TFAM overexpression is compared to the TFAM overexpression combined with ATF4 knock-down condition.

Figure 1G (now Figure 2D) shows a plot of the expression levels of the 22 genes that are significantly mis-regulated in both in the TFAM overexpression versus control condition, and the TFAM overexpression versus TFAM overexpression combined with ATF4 RNAi condition. The expression of all 22 genes displays opposite regulation (i.e. upregulated in TFAM overexpression versus control, but downregulated in the TFAM overexpression versus TFAM overexpression combined with ATF4 RNAi and vice versa) and therefore they were all considered to have had their expression "reversed" by ATF4 RNAi. We have now explained this in the legends of Figure 2 and Tables S1, S4 and S5 and in the Materials and Methods.

We considered genes to be significantly up- or downregulated if the fold-change was $\geq \pm 1.2$, the adjusted p value < 0.05 and the fragments per kilobase of transcript per million mapped reads (FPKM) ≥ 1 in at least one condition (as stated in the Materials and Methods).

2. Why do the authors shift between a motor neuron driver in Fig. 1 (D42-Gal4) and a pan-neuronal driver (nSyb-Gal4) for the rest of the studies?

-We shifted to nSyb-Gal4 for the transcriptomic and metabolomic studies so that we could use the whole CNS with gene expression/metabolic changes in all neurons, rather than having to isolate motor neurons. We then performed behavioural analyses with nSyb-Gal4 to be consistent with the transcriptomics and metabolomics. The phenotypes we obtain are consistent regardless of whether we use motor neuron or pan-neuronal Gal4 drivers. For example, we now show that TFAM overexpression causes activation of ATF4 using OK371-Gal4, D42-Gal4 and nSyb-Gal4 drivers (Figure 1A-D and Figure S1B, C).

3. *Figure S2 and its accompanying legend would both benefit from revision and annotation to make them easier to interpret.*

-We have revised and annotated Figure S2 and the legend to make them easier to interpret.

4. *A minor sentence error: on pg. 5, "The mitochondrial stress-induced activation of ATF4 is abolished by knock-down [of] PERK, but not GCN2..."*

-Thanks for spotting this error, we have corrected it.

Reviewer #2

a) The authors postulate that L-2-HG accumulates in response to ATF4 induction due to the promiscuous activity of LDH in neurons. In the literature, additional metabolic enzymes (i.e. MDH, PHGDH) have been reported to have a similar promiscuous activity. Are 3PG and malate levels altered upon ATF4 induction?

-Malate levels are not significantly different between control and TFAM overexpression or between control and TFAM overexpression combined with ATF4 knock-down conditions (malate levels shown in Table S6). 2/3-PG levels are significantly reduced by TFAM overexpression (Figure 3A) and are not significantly different between TFAM overexpression and TFAM overexpression combined with ATF4 knock-down conditions (2/3-PG levels shown in Table S6). Therefore, from our data there is no evidence that L-2-HG accumulates in neurons in response to ATF4 induction due to the activity of MDH or PHGDH.

b) What are the effects of LDH loss on 2HG production?

Drosophila Ldh null mutants for have a 98% reduction in L-2-HG levels (Li et al. 2017, *PNAS*, 114: 1353-1358, see Figure 2F). LDH is therefore essential for the production of L-2-HG in *Drosophila*.

c) How ATF4 regulates LDH-dependent 2HG production remains ill-defined. Is LDH expression altered upon ATF4 activation? Or does ATF4 induction promote cellular acidity or changes in redox?

-Our current data show that ATF4 activation is necessary for the increase in LDH expression and 2-HG levels caused by neuronal mitochondrial stress (Figure 2D and 3G). Consistent with our data, Lee et al., previously showed that ubiquitous knock-down of ATF4 in adult flies decreases LDH expression (Lee et al. 2015, *G3*, 5: 667–675, see Figure 7C). To address how ATF4 activation affects gene expression we have now included new transcriptomic data from CNS tissue overexpressing ATF4 in neurons (now shown in Figure 2B, C and Tables S2 and S3). ATF4 overexpression mis-regulates the expression of 149 genes (Figure 2B and Table S2). 24 genes are mis-regulated in both TFAM overexpression and ATF4 overexpression conditions and the expression of 22 of these genes changes in the same

direction in both conditions and are highly correlated (now shown in Figure 2B, C and Table S2 and S3), consistent with TFAM overexpression activating ATF4. LDH expression is significantly mis-regulated by ATF4 overexpression, confirming that LDH is a target of ATF4, but surprisingly its level is reduced by ATF4 overexpression (Table S2). This may be because when ATF4 is overexpressed alone it has different transcriptional activity to when it is activated as part of the mitochondrial stress response. TFAM overexpression activates the ER UPR (including XBP1, Figure 5I-K) as well as HIF-1alpha (as we showed in Cagin et al, 2015) and potentially other transcriptional regulators. These factors may interact with and modify the action of ATF4 causing it to promote LDH expression.

-To assess changes in redox we used Amplex red to analyse hydrogen peroxide levels in flies overexpressing TFAM and with ATF4 knock-down in neurons. Consistent with our previous study, where we showed that glutathione redox potential was significantly reduced in motor neurons overexpressing TFAM (Cagin et al., 2015), Amplex red analysis shows that TFAM overexpression causes a small but significant decrease in hydrogen peroxide levels. Overexpression of TFAM combined with ATF4 knock-down causes a similar reduction in ROS to TFAM overexpression alone. Therefore knock-down of ATF4 does not modify the reduction in ROS caused by mitochondrial dysfunction. These new data are shown in Figure S3P and described in the Results on p.6.

d) Since an ATF4 antibody was generated, the ATF4 knockdown in Figure S1C should be validated by western blotting in addition to the shown mRNA levels.

-Our ATF4 antibody does not recognise ATF4 on western blot, and so unfortunately we cannot perform this experiment. The antibody was generated against a soluble recombinant 60 amino acid fragment of *Drosophila* ATF4 and so likely only recognises ATF4 in its native conformation. However, the ATF4 RNAi we used has been shown by others to knock-down ATF4 expression in *Drosophila* (Lee et al. 2015, G3, 5: 667–675, see Figure 7B) and so we are confident about the efficacy of this RNAi.

e) The manuscript would be strengthened by further evidence that the in vivo phenotypes are connected to ATF.

-We have now included data showing that the pupal lethality caused by pan-neuronal TFAM overexpression with nSyb-Gal4 is suppressed by knock-down of ATF4 (Figure 1G). These data are consistent with the climbing and wing inflation data (Figure 1E, F) showing that the in vivo phenotypes caused by mitochondrial dysfunction are connected to ATF4. Note that pan-neuronal overexpression of TFAM with nSyb-Gal4 causes pupal lethality (as shown in Figure 1G) and we use nSybGal4 combined with heterozygosity for a TFAM mutation to recover viable adults overexpressing TFAM in the adult brain (as described on p.4 and the Materials and Methods). These new data are shown in Figure 1G and described in the Results on p.4.

Reviewer #3

1. In figure 1, the authors overexpressed TFAM in larval motor neurons using OK371-Gal4 (1A-C), and beautifully showed that this results in nuclear accumulation of ATF4. However, in 1D-E, they performed their locomotory assays in adult flies using a different Gal4 driver (D42-Gal4). Are there any locomotory assays the authors can perform on larvae to directly correlate the larval TFAM-dependent nuclear localization of ATF4 with a locomotory defect? Alternatively, nuclear localization of ATF4 should be shown in neurons overexpressing TFAM using the D42-Gal4 line. By using the same Gal4 line to show both phenotypes described, it will help directly correlate the locomotory defect with TFAM/ATF4 overexpression.

-Our experience of larval locomotory assays is that the phenotypes are highly variable and so we do not have robust data using these methods. As you suggest instead, we now show that overexpression of TFAM using D42-Gal4 causes nuclear localisation of ATF4 in larval motor neurons (Figure S1B, C). In addition, we now show that overexpression of TFAM with nSyb-Gal4 causes accumulation of ATF4 in the adult brain (Figure 1C, D in the revised manuscript). Our locomotory data with ATF4 knock-down are therefore consistent with the accumulation of ATF4. These new data are shown in Figures 1C, D and Figure S1B, C and described in the text on p.3.

2. Many of the figures involved pan-neuronal expression of transgenes using nSyb-Gal4. This seems reasonable as the paradigm used by the authors allowed the recovery of viable adults. The authors should repeat the experiments in Figure 4A-K with nSyb-Gal4, to rule out possible developmental defects giving rise to some of the phenotypes described in Figure 4. For instance, the increase in phospho-eIF2a in response to TFAM overexpression should be shown as a western blot on head extracts overexpressing TFAM with nSyb-Gal4, etc.

3. Similarly, use western blot on head extracts to resolve whether nSyb-Gal4 mediated overexpression of TFAM and PERK RNAi suppress the enhanced ATF4 expression evident when TFAM alone is overexpressed. Along the same lines can the enhanced GFP signal of XBPI-GFP be reproduced on western blots from head extracts?

-Our ATF4 antibody does not recognise ATF4 on western blot. The antibody was generated against a soluble recombinant 60 amino acid fragment of Drosophila ATF4 and so likely only recognises ATF4 in its native conformation. As mentioned above, we now show that overexpression of TFAM with nSyb-Gal4 causes accumulation of ATF4 in the adult brain (Figure 1D). However, nSyb-Gal4 mediated TFAM overexpression combined with PERK RNAi causes pupal lethality, so we cannot analyse ATF4 levels in the adult brain in this genotype by immunofluorescence.

-Unfortunately, nSyb-Gal4 mediated TFAM overexpression combined XBPI-GFP expression is pupal lethal so we cannot analyse XBPI-GFP levels in the adult brain in the genotype.

4. The authors explored the possibility that alterations in Calcium signaling could trigger some of the phenotypes observed. However, given the plethora of literature connecting redox signaling to neurodegeneration, ROS levels should also be quantified in the head extracts by the Amplex red assay to ascertain if ROS levels are affected as well. Its possible that 2-HG induces oxidative stress to cause neurodegeneration.

-To quantify ROS we used Amplex red to analyse hydrogen peroxide levels in head extracts from flies overexpressing TFAM alone and with ATF4 knock-down using nSyb-Gal4. Consistent with our previous study, where we showed that glutathione redox potential was significantly reduced in motor neurons overexpressing TFAM (Cagin et al., 2015), Amplex red analysis shows that TFAM overexpression causes a small but significant decrease in ROS levels. Overexpression of TFAM combined with ATF4 knock-down causes a similar reduction in ROS to TFAM overexpression alone. These data show that ATF4 (and likely 2-HG) do not induce oxidative stress to cause neurodegeneration. These new data are shown in Figure S3P and described in the results on p.6.

5. Is the increase in 2-hydroxyglutarate observed when TFAM is overexpressed a general feature of mitochondrial stress? Do other paradigms of mitochondria stress such as expression of RNAi to OXPHOS proteins or mitochondrial ribosomal proteins cause an increase in 2-hydroxyglutarate levels?

-We tried pan-neuronal knock-down of a number of OXPHOS proteins and mitochondrial ribosomal proteins but the RNAi lines that gave efficient knock-down caused larval or pupal lethality. We aim to test more RNAi lines in future as part of a larger effort to understand further how 2-HG regulates neuronal function in response to mitochondrial stress. However, there is already abundant evidence that increased 2-HG is a general feature of mitochondrial stress. Increased 2-HG has been observed in patients with OXPHOS complex I, III and multiple complex deficiencies and in mammalian cellular models of complex III deficiency and Leigh syndrome. In *Drosophila*, increased 2-HG has been observed in Pink1 mutant flies and in flies with a mutation in the mitochondrial citrate carrier. We describe these studies in the Discussion of the revised manuscript on p.7.

September 4, 2019

RE: JCB Manuscript #201904148R

Dr. Joseph Bateman
King's College London
125 Coldharbour lane
London SE5 9NU
United Kingdom

Dear Dr. Bateman,

Thank you for submitting your revised manuscript entitled "Mitochondrial stress-induced L-2-hydroxyglutarate regulates neuronal function". In light of the reviewer support during the first round of review and the degree of revision, we editorially assessed the changes you made to address Reviewers #1-2's points and consulted Reviewer #3. You will see that Rev#3 now recommends publication. We also feel that overall your responses appropriately resolve all the questions raised in review and we would be happy to publish your paper in JCB pending final revisions necessary to meet our formatting guidelines (see details below).

1) Titles, eTOC: Please consider the following revision suggestions aimed at increasing the accessibility of the work for a broad audience and non-experts.

Title: Mitochondrial stress causes neuronal dysfunction via an ATF4-dependent increase in L-2-hydroxyglutarate

(We find that adding the involvement of ATF4 is important and likely to appeal to a broad cell biology audience so would suggest editing the title to more fully reflect the advance)

eTOC summary: A 40-word summary that describes the context and significance of the findings for a general readership should be included on the title page. The statement should be written in the present tense and refer to the work in the third person.

- Please include a short eTOC statement on the title page of the resubmission.

It should start with "First author(s) et al..." to match our preferred style.

2) Reports contain an "Introduction" and a combined "Results and Discussion" section. Please be sure to title the sections appropriately.

3) Statistical analysis: Error bars on graphic representations of numerical data must be clearly described in the figure legend. The number of independent data points (n) represented in a graph must be indicated in the legend. Statistical methods should be explained in full in the materials and methods. For figures presenting pooled data the statistical measure should be defined in the figure legends.

Please indicate n/sample size/how many experiments the data are representative of: 1E, figure 3, 4ABDE, 5CHK, S1DF, S2, S3EJOP

4) Materials and methods: Should be comprehensive and not simply reference a previous publication for details on how an experiment was performed. Please provide full descriptions in the text for readers who may not have access to referenced manuscripts.

- More information about climbing and wing inflation assays, even if described in other work previously.

- Please provide more detail about the procedures used for tissue preparation for imaging, even if described in other published work.

- Microscope image acquisition: The following information must be provided about the acquisition and processing of images:

a. Make and model of microscope

b. Type, magnification, and numerical aperture of the objective lenses

c. Temperature

d. imaging medium

e. Fluorochromes

f. Camera make and model

g. Acquisition software

h. Any software used for image processing subsequent to data acquisition. Please include details and types of operations involved (e.g., type of deconvolution, 3D reconstitutions, surface or volume rendering, gamma adjustments, etc.).

5) A summary paragraph of all supplemental material should appear at the end of the Materials and methods section.

A. MANUSCRIPT ORGANIZATION AND FORMATTING:

Full guidelines are available on our Instructions for Authors page, <http://jcb.rupress.org/submission-guidelines#revised>. **Submission of a paper that does not conform to JCB guidelines will delay the acceptance of your manuscript.**

B. FINAL FILES:

-- High-resolution figure and video files: See our detailed guidelines for preparing your production-ready images, <http://jcb.rupress.org/fig-vid-guidelines>.

****The license to publish form must be signed before your manuscript can be sent to production. A link to the electronic license to publish form will be sent to the corresponding author only. Please take a moment to check your funder requirements before choosing the appropriate license.****

Thank you for this interesting contribution, we look forward to publishing your paper in the Journal of Cell Biology.

Sincerely,

Johan Auwerx, MD, PhD
Monitoring Editor, Journal of Cell Biology

Melina Casadio, PhD
Senior Scientific Editor, Journal of Cell Biology

Reviewer #3 (Comments to the Authors (Required)):

The authors have satisfactorily addressed all my critiques